



# Evaluation of Simulated Cloud Water in Low Clouds over the Beaufort Sea in Arctic System Reanalysis using ARISE Airborne In Situ Observations

J. Brant Dodson[1], Patrick C. Taylor[2], Richard H. Moore[2], David H. Bromwich[3], Keith M. Hines[3], Kenneth
L. Thornhill[1], Chelsea A. Corr[4], Bruce E. Anderson[2], Edward L. Winstead[1], and Joseph R. Bennett[5]

[1]Science Systems and Applications, Inc., Hampton, VA, 23666, USA
[2]NASA Langley Research Center, Hampton, VA, 23681-2199, USA
[3]Polar Meteorology Group, Byrd Polar & Climate Research Center, The Ohio State University, Columbus, Ohio, 43210, USA
[4]Department of Biology/Chemistry, Springfield College, Springfield, MA, 01109, USA
[5]National Suborbital Research Center, NASA AMES Research Center, Moffett Field, CA 94035-0001, USA

*Correspondence to*: J. Brant Dodson (jason.b.dodson@nasa.gov)

**Abstract.** Arctic low clouds and the water they contain influence the evolution of the Arctic system through their effects on radiative fluxes, boundary layer mixing, stability, turbulence, humidity, and precipitation. Atmospheric models struggle to accurately simulate the occurrence and properties of Arctic low clouds, stemming from errors in both the simulated atmospheric state and the dependence of cloud properties on the atmospheric state. Knowledge of the contributions from these two factors to the model errors allows for the isolation of the process contributions to the model-observation differences. We analyze the differences between the Arctic System Reanalysis version 2 (ASR) and data taken during the September 2014 Arctic Radiation-IceBridge Sea and Ice Experiment (ARISE) airborne campaign conducted over the Beaufort Sea. The results show that ASR produces less total and liquid cloud water than observed along the flight track and is unable to simulate observed large in-cloud water content. Contributing to this bias, ASR is warmer by nearly 1.5 K and drier by 0.06 g kg$^{-1}$ (relative humidity 4.3% lower) than observed. Moreover, ASR produces cloud water over a much narrower range of thermodynamic conditions than shown in ARISE observations. Analyzing the ARISE-ASR differences by thermodynamic conditions, our results indicate that the differences are primarily attributed to disagreements in the cloud-thermodynamic relationships, and secondarily (but importantly) to differences in the occurrence frequency of thermodynamic regimes. The ratio of the factors is about 2/3 to 1/3. Substantial sampling uncertainties are found within low likelihood atmospheric regimes; sampling noise cannot be ruled out as a cause of observation-model differences, despite large differences. Thus, an important lesson from this analysis is that when comparing in situ airborne data and model output, one should not restrict the comparison to flight track-only model output.



## 1 Introduction

The Arctic Ocean represents a critical component of the Arctic climate system. Moreover, the state and variability of the atmosphere, ocean, and sea ice in the central Arctic have significant implications for the climate across much of the planet (e.g., Deser et al. 2016). The complex interactions between surface properties, meteorological processes, and the surface energy budget are critical to understanding the climate of the Arctic Ocean and making projections of its response to anthropogenic climate change (e.g., Budyko 1969; Manabe and Wetherald 1975; Hall 2004; Boeke and Taylor 2018; Dai et al. 2019). Clouds

represent a key feature of the Arctic climate system and modulate important aspects of ice-atmosphere-ocean coupling processes that strongly influence the evolution of the Arctic Ocean climate (Curry et al. 1996).

The effects of clouds on the Arctic climate system span a wide range of physical processes, such as heating and cooling via cloud radiative effects (CREs), mixing of the boundary layer by generating turbulence, influencing humidity through precipitation, and altering boundary layer stability (Vihma et al. 2014). Through these processes, clouds shape the temperature

and sea ice variability and trends in the central Arctic representing a potentially significant climate feedback. Cloud-radiative feedbacks, in particular, remain a key uncertainty in the Arctic climate system (Curry et al. 1996; Wendisch et al. 2018).

Models differ strongly in their simulation of CREs (Karlsson and Svensson 2011; English et al. 2015; Boeke and Taylor 2016) and in their representation of the Arctic cloud annual cycle (Taylor et al. 2019). Systematic biases in CREs impact the representation of atmospheric circulation variability and the lifetime of atmospheric circulation regimes (Stramler et al. 2011;

Pithan et al. 2014). Moreover, CREs influence sea ice anomalies during both melt and growth seasons (Kay et al. 2008; Hegyi and Taylor 2018) and reduce shortwave downwelling radiation at the surface (Sledd and L'Ecuyer 2019; Alkama et al. 2020). Sledd and L'Ecuyer (2019) quantify the cloud masking effect on the sea ice albedo perturbation at TOA to be ~50%. Moreover, Alkama et al. (2020) argue that accurate representation of present-day Arctic cloud properties is necessary for climate models to accurately reproduce the surface radiation budget impact of sea ice variability and change. Thus, accurate representation of

Arctic cloud properties in atmospheric models is a necessary but insufficient condition to ensure accurate predictions of Arctic climate change. The understanding of the key processes that give rise to the mean and variability of cloud properties within atmospheric regimes is lacking; this knowledge is needed in order to understand the role of clouds in the atmospheric circulation, cloud feedback, and improve the simulations of clouds within models.

A process-oriented approach that moves beyond the analysis of monthly mean properties is required to identify and understand

the processes that cause differences between observations and the model representation of clouds. Previous analyses have identified a range of errors in the model simulations at monthly or daily mean scales (e.g., Karlsson and Svensson 2011; English et al. 2015; Boeke and Taylor 2016). However, these approaches fail to unequivocally identify the processes responsible for the errors. Detailed observations of cloud properties and the meteorological environments are needed to identify errors in the simulation of Arctic clouds. However, these observations are difficult to obtain from satellites – cloud water

retrievals from passive instruments is challenging (Shupe et al. 2005; Ehrlich et al. 2017; Chen et al. 2020) and the cloud presence limits the ability of sounding instruments to sample the thermodynamic properties of the underlying boundary layer.



To fill these data gaps, aircraft field campaigns are necessary to collect the unavailable data *in situ*. One such campaign was the Arctic Radiation - IceBridge Sea and Ice Experiment (ARISE), conducted in the Beaufort Sea region during September 2014 (Smith et al. 2017). In this study, we take a process-oriented approach (e.g., Taylor et al. 2019) by investigating

dependence of *in situ* cloud properties on varying meteorological regimes, and use these results to evaluate the Arctic System Reanalysis version 2 (ASR; Bromwich et al. 2018).

The main purpose of this paper is to document the properties of Arctic boundary layer clouds, their meteorological environments, and their relationships, using ARISE observations and compare these results with ASR, a reanalysis specifically designed to represent Arctic weather and climate (Bromwich et al. 2018). Model-observation disagreements occur for at least

three general reasons. First, the model may produce incorrect meteorological conditions, such as biases in the mean or variability of temperature or humidity. Second, the model may misrepresent the observed relationships between cloud and meteorological conditions. For example, the model may produce too much or too little cloud water at a given humidity value even when the humidity is similar to the observed value. Lastly, flight track sampling can limit precision and range of conditions over which relationships between clouds and meteorological conditions can be determined. We assess the impact

of the flight track to sampling on the ability to assess model realism using flight campaign data. Our analysis addresses three questions:

(1) How do the cloud properties in ARISE and ASR compare?

(2) Can discrepancies between ARISE and ASR be explained by grid-scale disagreements in meteorological conditions and/or errors in the simulated cloud/thermodynamic relationships?

(3) How are ARISE/ASR comparisons affected by flight track sampling?

Section 2 discusses the observational and ASR datasets, as well as the analysis methodology. Section 3 examines flight track cloud properties observed from the aircraft and ASR output. Section 4 discusses the relationships between cloud properties and thermodynamic regimes in the *in situ* and ASR. Section 5 presents the quantification of the effects of sampling and Section 6 presents our conclusions.

## 2 Data and Model

### 2.1 The Arctic Radiation - IceBridge Sea and Ice Experiment

ARISE was conducted during September 2014 to investigate the radiative properties of the Arctic and the interconnections with meteorological conditions (Smith et al. 2017). Measurements of cloud properties, radiative fluxes, and other meteorological variables were carried out by a NASA Wallops Flight Facility C-130 aircraft based at Eielson Air Force Base

near Fairbanks, AK. The observing region encompassed a portion of the Beaufort Sea north of Alaska (Fig. 1) that contained both sea ice and ice-free ocean. For flights used in this study, the aircraft typically entered and exited the observing region flying well above the boundary layer then descended and spent most of the time below 1000 m adjusting its altitude to fly above, below, and within the cloud layer.





ARISE consisted of 16 total flights over the Beaufort Sea; however, only a subset of those are used in this study. For a flight
to be included in the analysis, it must have collected cloud and thermodynamic property measurements below 1000 m altitude
for at least an hour. Eight flights met this requirement: September 13, 15-19, 21, and 24. As an example, Fig. 1 shows the
science flight and cloud probe data from September 15, 2014. Only aircraft observations taken over sea ice and ocean below
1000 m altitude are shown to focus on properties of low clouds over the Arctic Ocean. Additionally, only observations that
report thermodynamic and cloud microphysical variables simultaneously are included – cases where one is missing are rare.
The measurements of interest include total cloud water (cloud liquid + cloud ice), air temperature and humidity. Cloud water
measurements were collected by the WCM-2000 probe (Smith et al. 2017; Fig. 1), which reports total and liquid water content;
ice water content is derived as the difference. The WCM-2000 probe reports two liquid water content values obtained from
wires of different diameter: 2.0 and 0.5 mm. We convert these measurements to total cloud water (QC), cloud liquid (QL), and
cloud ice (QI) specific humidity for comparison with ASR. Flight-level *in situ* static air temperature (T), relative humidity with
respect to liquid (RH), and specific humidity of vapor (QV) are also used.
The WCM-2000 data are used in this study and represent the most "direct" measurements of QC/QL/QI. As an additional
check of the robustness of WCM-2000 measurements, cloud droplet volume concentration data from the aircraft-mounted
Cloud Droplet Probe (CDP) are integrated to compute QC. Figure 2 shows that WCM-2000 detects more QC than the CDP
especially in clouds with large QC values. This holds true for QL values measured by both wire diameters. Overall, WCM-
2000 QC agrees well with the integrated CDP QC with a substantial positive correlation (0.85). The LWC derived from the
two different diameter wires also show substantial correlation (~0.8). Given this consistency, we proceed using QL determined
from the 2.0 mm wire, as the larger diameter captures more liquid water.

## 2.2 Arctic System Reanalysis version 2 (ASR)

The Arctic System Reanalysis is a regional reanalysis of the greater Arctic stretching from the headwaters of the major
northward flowing rivers to the North Pole and is targeted at producing a refined description of the Arctic environment during
the current period of rapid change (Bromwich et al. 2016, 2018). It is based upon the polar version of the Weather Research
and Forecasting model(known as Polar WRF; http://polarmet.osu.edu/PWRF/) that features realistic heat transfer through
snow and ice, comprehensive description of sea ice characteristics, and other improvements to the Noah Land Surface Model.
The model physical parameterizations are chosen for Arctic conditions based upon extensive testing (e.g., Bromwich et al.
2009). In particular, the Goddard microphysics scheme (Tao and Simpson 1993; Tao et al. 2003) is used to predict the cloud
species studied here; this is now a dated scheme in light of the recent advances in microphysics research that have emphasized
prediction of cloud liquid water at low air temperatures (e.g., Hines et al. 2019). ASR assimilates a wide variety of conventional
meteorological observations as well as satellite-derived radiances and retrievals using three-dimensional data assimilation.
Specified sea ice conditions, snow cover and vegetation characteristics are updated either daily (sea ice) or weekly. ASR is
produced every three hours on 71 vertical model levels with 25 below 850 hPa to better resolve the highly stratified near-



surface conditions often present in the Arctic. In contrast to version 1, ASR version 2 (NCAR, 2017) has higher spatial resolution (15 km) and implemented sub-grid (convective) cloud fraction over land to improve the surface downward radiative fluxes. Recent investigations have demonstrated ASRv2 skill in capturing a variety of Arctic phenomena, such as snowfall over the Arctic Ocean (Edel et al. 2020) and polar lows (Stoll et al. 2018). Currently, ASRv2 spans 2000-2016 coinciding with

the availability of the essential input data sets from MODIS and is being updated through 2020. A new version (ASRv3) is being developed to deal with issues such as explored here.

**2.3 ARISE and ASR Collocation Methodology**

ARISE is conformed to the ASR spatial grid and three-hour resolution as follows:

(1) Each ARISE data point (one per second) collected at or below 1000 m altitude is collocated with the center of the nearest

ASR grid box.

(2) Consecutive sets of ARISE data points within the same ASR grid box are then collocated with the nearest ASR vertical level.

(3) Each consecutive set of ARISE data within the same grid box and altitude are averaged.

(4) The median time of observation for each consecutive set is chosen to represent the entire set.

(5) Each set of data points is collocated with the nearest ASR time step – flights lasting longer than three hours and different flight segments are collocated with different time steps.

Collocating ARISE data with an ASR grid box allows the simulated values of cloud and thermodynamic properties to be directly compared, reducing the noise from high frequency ARISE sampling (which would be subgrid-scale artifacts compared with the ASR grid). Organizing the ARISE and ASR data in this way allows an optimized comparison and minimizes the

possible sampling effects.

**3 Mean cloud and thermodynamic properties from ARISE and ASR**

**3.1 Probability density functions of cloud and thermodynamic variables**

An evaluation of ASR-simulated Arctic thermodynamic and cloud properties is performed by comparing the probability density functions (PDFs) of QC and thermodynamic variables. These variables are divided into discrete thermodynamic

regimes separated by 5 °C in T, 1 g kg⁻¹ in QV, and 10% in RH. Figure 3 shows PDFs of the ARISE and ASR variables along the flight track. First, Fig. 3a indicates that ASR simulates less QC than observed, manifesting as fewer occurrences of high QC values. When restraining the ASR data to the ARISE flight track (clear and cloudy regions), ASR produces 21% of the observed average total QC (0.007 g kg⁻¹ versus 0.033 g kg⁻¹). When expanded to the full Beaufort Sea, ASR produces 0.018 g kg⁻¹ (56% of ARISE). ASR significantly underrepresents QC in low Arctic clouds.

The thermodynamic regime PDFs offer clues as to why ASR has less QC. Considering all conditions (clear and cloudy) along the ARISE flight track, ASR is on average 1.40 K warmer with a lower QV (0.06 g kg⁻¹) than observed, resulting in an 4.3%



lower average RH. The reduced RH inhibits cloud formation and thus the generation of QC in ASR. The warm and dry biases across the full Beaufort Sea domain are even greater. Segal-Rozenhaimer et al. (2018) found similar warm and dry biases in MERRA-2 relative to ARISE, suggesting that the warm bias in the lower Arctic troposphere may be a common feature of

meteorological reanalysis.

Considering only cloudy grid boxes (defined as total QC > 0.02 g kg$^{-1}$; Fig. 3e), ASR average QC along the ARISE flight tracks is 55% of the observed value (0.06 g kg$^{-1}$ versus 0.11 g kg$^{-1}$, respectively). Because the cloud-only QC value for ASR is twice that of the clear + cloudy value, the lack of QC in ASR stems from both less frequent cloudy grid boxes and smaller in-cloud QC values. In-cloud flight track ASR T is 3.10 K warmer and QV is 0.45 g kg$^{-1}$ greater than observed, with similar

values for the whole Beaufort Sea. The higher QV partially offsets the drying effect of warmer T on RH, resulting in RH for ASR being 5.8% greater. Despite the higher QV, the QC PDF in cloudy grid boxes (Fig. 3e) shows that ASR still produces large QC values less frequently than observed. This suggests that the high QV is not converted to QC by the parameterized cloud physics as efficiently as observed, pointing to a key deficiency.

One additional feature of note in Fig. 3h, ASR rarely produces clouds in grid boxes with less than 90% RH, while ARISE data

show clouds occurring (QC < 0.02 g kg$^{-1}$) at grid box averaged RH values as low as 40%. ASR is coded to cap grid box RH at 100%, and also to not produce clouds below 80% RH, which (mostly) prohibits the presence of clouds in marginally humid conditions. Thus, ASR forms clouds over a narrower range of atmospheric conditions (specifically, RH values) than observed *in situ*.

## 3.2 Vertical cloud water and thermodynamic profiles

Figure 4 shows the vertical distribution of QC in ARISE and ASR below 1000 m. Observed QC shows two primary maxima, ~400 m and at or above 1000 m; these maxima also align with QL profiles over the Arctic Ocean from CALIPSO-CloudSat data (e.g., Taylor et al. 2015). Examining cloudy grid boxes, the QC maximum near 1000 m is larger than that at ~400 m. Both maxima are dominated by the liquid phase.

ASR shows multiple deficiencies in the simulated QC vertical profile compared with ARISE. These deficiencies are more

evident when comparing along the ARISE flight tracks but are also found when considering the Beaufort Sea domain (Fig. 4c,f). Most notable, ASR produces too little QC above 500 m, to the point that the higher altitude maximum is almost non-existent. Within cloudy grid boxes, ASR reproduces the amplitude of the lower altitude maximum, up to 300 m. A major component of the disagreement likely stems from a poor representation of the processes that generate the upper cloud layer, including interactions with large-scale advection and the free troposphere (e.g., Shupe et al. 2013). Additionally, ASR

simulates very little QI during the campaign, indicating that mixed-phase cloud microphysics are not a factor driving the lack of QC in ASR. Moreover, Segal-Rozenhaimer et al. (2018) also identified a lack of cloud cover in MERRA-2 relative to ARISE data, which also corresponds with warm/dry biases. However, they reported missing clouds below 500 m, whereas we find missing clouds (in the form of less QC) above 500 m.





The partitioning between QC and precipitation represent a possible reason for the ASR-ARISE difference; ARISE observations
do not partition between QC and precipitation whereas the ASR bulk cloud microphysics does. Thus, the missing QC in ASR
could result from an overactive precipitation conversion. To investigate this, Figs. 4b-c and 4e-f plot the sum of QC and
precipitation (dashed line) in addition to QC. Precipitation water more than doubles the total condensate above 500 m and yet
this does not bring the QC close to the observed value. Thus, there is no evidence that an overactive conversion parameter
explains the lack of QC above 500 m.

Because of the close interplay of clouds and thermodynamics, the errors in the ASR vertical cloud profile are likely related to
errors in the boundary layer thermodynamic structure. Figure 5 shows the T, QV, and RH vertical profiles for ARISE and
ASR. ASR has a greater T and lower RH than ARISE, while QV is similar. This corresponds with the PDFs shown in Fig. 3
and demonstrates that the warm bias in ASR is the main contributor to the low RH. The warm and dry ASR biases are also
found across the Beaufort Sea domain (not shown), demonstrating that the thermodynamic errors are not statistical artifacts
from the flight path sampling.

The warm and dry biases extend throughout the boundary layer, with no obvious altitude dependence. This is different than
the disagreement reported by Segal-Rozenhaimer et al. (2018), where the warm and dry biases in MERRA-2 are mostly
confined to below 500 m. There is likely a connection between the lack of cloud cover and the warm/dry biases in MERRA-2
below 500 m, whereas there is no obvious direct connection between the lack of QC above 500 m and the warm/dry bias
extending evenly throughout the boundary layer. Therefore, we speculate that the MERRA-2 and ASR cloud vertical
distributions differences with observations have different causes.

## 4 Comparison of cloud water versus thermodynamic variables

### 4.1 Bivariate analysis

The T and RH biases in ASR represent one type of contribution to the low mean QC bias in ASR. However, the relationships
between QC and thermodynamic variables are another important contributor. Figure 6 shows the observed and simulated
bivariate relationships between QC and T, QV, and RH. ARISE data shows increased QC at higher RH, as expected, but also
at low T and QV. The latter relationship is curious, as higher QV provides more vapor for conversion into QC; however, the
small correlation (0.23) indicates a weak relationship. This is not a result of an unusual sensitivity of RH to T and QV in the
observations and ASR, as the sensitivities in ARISE and ASR are almost identical. Compared with ARISE, ASR also shows
QC increasing with RH (Fig. 6). As the PDFs in Fig. 3 suggest, ASR produces very little QC when RH is below 90%. These
results also hold when considering cloudy grid boxes (Fig. 7). ASR has difficulty producing QC outside of an unrealistically
narrow range of meteorological conditions, contributing to the low mean QC bias. These simulated dependencies of QC on T
and QV disagree with observations influencing the low QC bias in ASR.

Unlike the ARISE results, ASR information is not limited to the ARISE flight path. It is possible to look at the simulated QC-
thermodynamic variable relationships considering all ASR grid boxes within the Beaufort Sea domain during ARISE. This is





useful for characterizing the simulated cloud-thermodynamics relationships more completely, such that they are not affected by the limited flight track sampling. To determine the extent to which the ASR results are affected by the flight track sampling, Fig. 6g-i show the relationships in ASR when considering the Beaufort Sea domain. Compared to Fig 6d-f, the full dataset shows a larger sensitivity of QC to T, and also a few cases in which cloudy grid boxes occur with RH below 90%. However,

the unrealistic relationship between QC and QV persists.

## 4.2 Contributions of error types to ASR cloud water

Taken together, Figs. 3 and 6 qualitatively show that the deficiency in ASR QC is related to errors in the simulated thermodynamic conditions and errors in the QC-thermodynamics relationships. We quantify the contributions of these errors sources to the low QC bias by constructing a synthetic QC data set from the observed thermodynamic conditions and ASR-

derived relationships between QC and thermodynamics. We then calculate the change in QC that occurs when the ASR thermodynamics and relationships are assumed to be "reality". Ideally ASR would produce 100% of the observed value; the closer a QC percentage is to 100%, the greater the improvement to ASR the change produces. The synthetic QC data set is constructed by substituting the ASR-derived relationship (from Fig. 6) for the observed ARISE QC values (e.g., for every instance that ARISE reported a T of approximately -10 °C, replace the observed ARISE QC value with 0.01 g kg$^{-1}$ from Fig.

6d). A mathematically equivalent way of doing this calculation is to multiply the ARISE PDF of T (Fig. 3b, black bars) by the ASR bivariate relationship of QC versus T (Fig. 6d, black curve). Averaging this product gives a QC value of 0.008 g kg$^{-1}$, 24% of the observed ARISE mean QC value of 0.033 g kg$^{-1}$. This shows that much of the QC deficit in ASR is related to ASR not producing enough QC when T is low. Similar results are found when considering different thermodynamic regime sized (e.g. T regime size are reduced from 5 °C to 2 °C).

To find the effect of the simulated ASR T values on the QC deficit, the converse calculation is performed - multiplying the ASR PDF of T (Fig. 3b, red bars) by the ARISE bivariate relationship between QC and T (Fig. 6a, black curve). Averaging this product yields a QC value of 0.014 g kg$^{-1}$ (45%). The larger resulting percentage shows that the warm bias in ASR produces significant, while slightly smaller, reduction in QC as the unrealistic QC-T relationship.

Performing the same calculations with QV tells a very different story. Substituting the ASR bivariate QC-QV relationship

results in a mean QC of 0.008 g kg$^{-1}$ (24%), a reduction roughly equivalent to the total ASR QC deficiency. However, using the ASR QV PDF and the observed QC-QV relationship gives a mean QC of 0.033, a 3% increase over the observed QC value. For QV, the erroneous bivariate relationship is the sole factor in the ASR QC deficiency. In other words, plenty of QV is available in the ASR atmosphere to produce the observed QC values; however, ASR fails to convert enough vapor into QC.

Performing the same calculations for RH produces mean QC values of 0.007 (23% of the ARISE mean QC) and 0.025 g kg$^{-1}$

(79%) when subbing the ASR QC-RH relationship and the RH PDF, respectively. In this case, ASR does not produce large RH values frequently enough and does not produce enough QC when RH is sufficiently large; however, the latter is primarily responsible for the small QC in ASR. Note that this analysis treats the three thermodynamic variables independently, whereas in reality they are interdependent – particularly RH on both T and QV. This is why the errors in simulated T and RH PDFs




both have such notable effects on the mean QC, and why the mean QC biases caused by the PDF biases and QC-thermodynamic
relationships do not cleanly sum up to 100%.

Switching the ASR output from the flight track-only to the Beaufort Sea domain shows the effects of the sampling and more complete understanding of which factors are responsible for the larger mean QC value (0.018 g kg$^{-1}$) for the Beaufort Sea Domain. The mean QC calculated when subbing the ASR QC-T relationship and the T PDF are 0.022 g kg$^{-1}$ (69%) and 0.013 g kg$^{-1}$ (41%), respectively. The more realistic QC-T relationship seen in Fig. 6g corresponds with the much larger ASR QC in
the full dataset, but the warm bias still results in a severely low ASR QC. The mean QC when subbing ASR QV and QC-QV relationship PDFs are 0.020 g kg$^{-1}$ (62%) and 0.025 g kg$^{-1}$ (77%). Thus, considering QC-QV relationship derived using the ASR output for the Beaufort Sea Domain is not as strong of a factor for reducing QC in the full ASR dataset compared with the flight track output. The larger QC value for 2 g kg$^{-1}$ QV in Fig. 6h versus Fig. 6e seems to be the most important factor in the larger mean QC in the full dataset.

Subbing RH has a very peculiar result: 0.013 g kg$^{-1}$ (39%) and 0.037 g kg$^{-1}$ (116%), which is strange considering that the domain-averaged ASR is drier than the flight tracks-averaged data. How does a drier atmosphere lead to increased QC? The answer lies in the RH PDF (Fig. 3h). Unlike ARISE and flight track ASR, the full Beaufort Sea ASR PDF is bimodal. The dry mode is responsible for the small mean ASR. The wet mode peaks at 100% RH, unlike the other two PDFs, and at the value for which observed QC (Fig. 6c) maximizes. The boost to QC given by the wet mode peak is overcomes the drying effect of
the dry mode. Thus, the change in the PDF shape of Beaufort Sea ASR RH from flight track PDFs leads to increased QC.

**4.3 Trivariate analysis using two-variable thermodynamic regimes**

The previous results warrant further trivariate analysis of the QC-thermodynamic relationships using two thermodynamic variables versus QC. Figure 8 shows that the observed QC is largest in conditions with high RH and low T (Fig. 8a), high RH and low QV (Fig. 8d), and low T and low QV (Fig. 8g). Together, QC during ARISE is largest in low T, low QV, and high
RH conditions. This result clarifies the curious relationship between smaller QV values and larger QC from Fig. 6b, as QV is constrained by the Clausius-Clapeyron relationship and must be small at low T, apparent in Fig. 8g. Moreover, ARISE data indicates a strong sensitivity of QC to T; a relationship that is much weaker in ASR.

The dependence of QC on the thermodynamic regimes in ASR along the flight track hardly resembles the observations (Fig. 8b,e,h). The lack of total QC in ASR is indicated as no values above 64 mg kg$^{-1}$ appear in Fig. 8b. The largest disagreement
between ARISE and ASR involves the observed relationship of large QC values at low T and low QV. Some of the disagreement stems from ASR not capturing all of the observed meteorological regimes. Particularly, the lowest T values did not occur in ASR along the flight path, illustrated by the gray boxes in Fig. 8h around -15°C. The dependence of QC in ASR shows that slightly larger QC values occur at lower T and large QV (Fig. 8h); a much weaker dependence than observed (Fig. 8g vs. Fig. 8h). Moreover, Fig. 8e indicates no preference for ASR to simulate large QC values at lower QV.

The narrower range of conditions sampled along the flight track within ASR again motivated an examination of the thermodynamic regimes within the larger Beaufort Sea domain. The comparison between the flight-track only and Beaufort





Sea domain results raises questions about the representativeness of the relationships found when using the flight track-only output. Figure panels 8c, 8f, and 8i show the QC-thermodynamic regimes determined for the Beaufort Sea domain. Comparing these results with panels Fig. 8b, 8e, and 8h reveals the larger range of thermodynamic conditions sampled (i.e. the non-gray regions are larger) enabling an assessment of ASR-simulated QC over a wider range of thermodynamic regimes. Figure 8i indicates that high QC values in regimes of low T/low QV manifest within the Beaufort Sea domain. Thus, at least some of the "missing" QC results from the flight track not intersecting the low T/low QV regime, produced nearby.

The simulated dependence of QC on thermodynamic regimes also contributes to the differences in the mean QC with ARISE. Specifically, ASR concentrates the largest QC values at the lowest ranges of T and QV showing a faster decrease in QC than ARISE as T and QV increase (about 40% faster at T of -10 °C and QV of 1 g kg$^{-1}$). ASR only produces large QC under the extreme ranges of thermodynamic. Thus, the inability of ASR to simulate larger QC under typical thermodynamic conditions contributes to the disagreements with ARISE.

Another noteworthy result is that the ASR Beaufort Sea domain analysis suggests a secondary regime of high T/high QV with large QC values. This high QC regime is not found in ARISE data. This result raises important questions about the realistic nature of this thermodynamic regime. Further exploration of this is beyond the scope of the present analysis.

## 4.4 Effect of joint reanalysis error types on simulated cloud water

The analysis discussed in subsection 4.2 is applied to quantify and compare the dependence of the ASR QC bias based upon the thermodynamic regime joint PDFs. Subbing the flight track ASR QC-thermodynamic regime relationships (Figs. 8b, 8e, and 8h) yields mean QC values of 0.007 g kg$^{-1}$ (23% of the ARISE mean QC) for RH and T, 0.006 g kg$^{-1}$ (19%) for RH and QV, and 0.008 g kg$^{-1}$ (25%) for T and QV. Subbing the ASR thermodynamic regime frequency of occurrence (not shown) results in mean QC values of 0.013 g kg$^{-1}$ (39%), 0.016 g kg$^{-1}$ (50%), and 0.015 g kg$^{-1}$ (45%). Thus, both the ASR QC-thermodynamic regime relationships and the differences in thermodynamic regime frequency contribute substantially to the low QC bias in ASR, though the former has the greater influence.

Subbing the full ASR dataset instead produces modest increases in mean QC. When substituting the QC-thermodynamic regime relationships, only the T-QV regime shows a significant increase in mean QC, 0.017 g kg$^{-1}$ (52%). This is connected with the improved representation of the observed large QC values with low T and QV. However, this is not sufficient to overcome the ASR's aversion to producing clouds in lower RH conditions. When subbing the ASR regime frequency of occurrence, RH and QV show the largest increase in mean QC, of 0.017 g kg$^{-1}$ (52%). The biases in thermodynamics lead to major underestimates of QC even when the full ASR dataset is used – adding more data does not resolve the deeply-rooted errors in T and RH. Of the three variable pairings, given the imperfect separation between these semi-dependent variables, improvement in mean QC comes primarily from the more realistic representation of the QC with T and QV relationship and the QC boosting effect of the RH PDF shape (Section 4.2).



## 5 Effect of sampling frequency on results

The previous section shows that a direct comparison of ARISE and ASR data along the ARISE flight path finds that the
relationships between cloud variables and thermodynamic regimes are substantially different. However, when considering
between QC-thermodynamic regime relationships in ASR for the entire Beaufort Sea domain and time period, the relationships
are established over a wider range of conditions and better resemble the ARISE data. Implying that a possible source of
uncertainty originates from the limited sampling of meteorological conditions along the flight path. At first glance this may be
an unintuitive result – shouldn't the along-track ASR results agree more closely with ARISE than the domain-average results?
But consider that the ASR-ARISE discrepancies in mean QC are partially understood by differences in the QC-thermodynamic
relationships and the differences between the sampled vs. simulated thermodynamic regimes; however, there is still the
question as to the magnitude of the sampling uncertainty in the mean QC. While it is not possible from ARISE data to quantify
the sampling uncertainty, it can be estimated using ASR output. In addition to quantifying sampling uncertainty, the knowledge
gained has utility for the design of future aircraft campaigns where observation-model comparison is a major goal.

### 5.1 Experiment setup

This experiment tests the effect of sampling on the ASR QC-thermodynamic regime relationship results by simulating alternate
ARISE aircraft flight paths and times within the operation domain. ARISE flight paths were constrained by the overpass times
of CERES-bearing satellites and not driven by meteorological conditions. This experiment uses a random sampling approach
to simulate alternate ARISE aircraft flight paths, assuming no targeting of any particular meteorological conditions to quantify
how the ASR-ARISE comparisons are affected by different flight configurations. The experiment proceeds as follows:
The eight flight paths during the low-level portions of the flights are randomly assigned days from the eight days that ARISE
flew, and random start times during the day.
Each randomized flight has a random starting point within the ARISE domain and a random rotation from the original flight
path orientation. Flight paths must stay within the domain at all times and those that leave are re-randomized.
ASR-simulated cloud and thermodynamic data are collected and processed into the QC-thermodynamic regime format as seen
in Fig. 8 and stored.
Steps 1-3 are repeated with each iteration of eight flights considered a "trial" campaign. After 50,000 trials are completed, the
results are used to tabulate indices that represent sampling uncertainty for each meteorological regime.

### 5.2 Results

Figure 9 displays three indices summarizing the sampling experiment results. Figure 9a, 9e, and 9i show the likelihood that
for a single experiment of eight flights that each meteorological regime is encountered at least once. The highest likelihood
lies near the center of the data range, as expected, though RH (QV) chances are slightly shifted towards higher (lower) values.
For meteorological regimes with large QC values, those with average T and lower QV are most likely to be sampled. Sampling



of the low T/low QV and high T/high QV regimes show a likelihood of 10% or less. As these regimes have the highest QC

values in the full dataset, this indicates that the ASR domain-averaged QC value is lower because the simulated aircraft has

the lowest probably of intercepting the extreme ranges of the thermodynamic regimes where largest QC values occur (assuming

the high T/high QV regime is realistic).

Figure panels 9b, 9f, and 9j show the number of samples collected in each meteorological regime on average across the

experiments. The number of samples is important for calculating reliable statistics for the QC values in each thermodynamic

regime. The results follow a similar pattern as the sampled likelihood results and the regimes with the largest QC have few if

any samples. Low T/low QV conditions are rare in ASR, less than 0.1% of all samples. This partially explains why the large

QC values for low T/low QV are not apparent along the flight track (Fig. 8h) even in the regimes that have data for both ASR

and ARISE.

Figures 9c, 9g, and 9k display the mean absolute error (MAE) of the QC for each thermodynamic regime. The MAE is

calculated as the average absolute value of the difference between each trial QC and the population QC (Figs. 9d, 9h, and 9l).

This metric quantifies how close to the population value a single trial's value can be expected to be for each thermodynamic

regime that is sampled in the trial – while the mean error would approach zero because of positive and negative differences,

the MAE does not and is similar to standard deviation. The magnitude of MAE scales closely with the population mean QC

value. In thermodynamic regimes with high QC values, MAE is typically about half of the population mean, meaning that

errors of ~50% can be expected for a trial experiment. However, this is not universal, particularly for the T and QV results,

where for example, the MAE at -10°C and 1 g kg$^{-1}$ (Fig. 9k) is 184% the population mean (8.9 mg kg$^{-1}$ versus 4.8 mg kg$^{-1}$).

Recall that this regime shows major disagreement between ARISE and ASR along the flight path (Figs. 8g and 8h) under low

T/low QV conditions. The large experimental MAE result suggests that a single trial is likely to miss the large QC value in

this regime even if it is sampled. In other words, these conditions can be associated with both cloudy and clear-sky conditions

in the ASR realizations. A similar issue is encountered the high T/high QV regime.

The standard deviations of the total QC across the experiments are shown in Figs. 9d, 9h, and 9l. The results resemble MAE,

with the notable exception of high T/high QV conditions. Because standard deviation tends to be more sensitive to outliers

than MAE, this result indicates that most random samplings of these conditions have an error smaller than the population mean

value with a small chance that a big outlier might be sampled instead. In other words, it is possible but unlikely to sample

clear-sky conditions in the high T/high QV regime on the flight track, and if this is the only sample, conclude that no clouds

occur under these conditions.

## 5.3 Implication of the sampling results

The random sampling approach is a powerful tool helping to assess how representative a single field campaign may be in

characterizing the cloud conditions within in each regime. The large MAE and standard deviations in some thermodynamic

regimes suggest that the relationships derived from flight track only information contain significant sampling uncertainty.



Based upon the ASR Beaufort Sea domain analysis, it seems serendipitous that the ARISE flights flew though regions of low T/low QV and captured the large QC values under these conditions. On the other hand, this serendipitous occurrence could have resulted from the cloud-seeking priority of the ARISE aircraft. Conversely, it could have been bad luck that ARISE missed high T/high QV cloudy conditions. Are the ARISE results a possible realization of the full ASR dataset? Figure 10

provides a method of assessing the representativeness of ARISE from the full ASR dataset. The majority of thermodynamic regimes with large QC values in both ARISE and ASR show that the ARISE values are within the sampling uncertainty estimated from the random sampling test. For the most part the ARISE data are a possible realization of the ASR dataset. Only ARISE-ASR QC differences at RH < 80% differ by more than two standard deviations.

More comprehensive evaluations using random sampling applied to satellite data may aid in future design of aircraft campaigns

to minimize that chances of missing meteorologically interesting conditions. Ideally, this could be accomplished by having the aircraft target conditions that *should* produce clouds according to the models rather than just seeking where there *are* clouds. As this may be challenging to forecast, other strategies such as increasing the number/duration of flights is a useful substitute.

## 6 Summary and Conclusions

We have examined the ability of ASR to replicate the observed cloud water, thermodynamic conditions, and the relationships

between them with respect to the ARISE aircraft field campaign observations during September 2014. This includes quantifying the sensitivity of cloud water to the thermodynamic regimes using methods frequently employed in studies of satellite data. Further, a Monte Carlo-style random sampling of the ASR output was employed to understand the influence of limited aircraft sampling on the results.

The results address the following questions:

*How does cloud water in ARISE and ASR compare?*

Compared with ARISE observations, ASR produces too little cloud water, about 25-50% of the observed value. Notably, ASR produces almost no cloud water above 500 m, whereas ARISE depicts cloud occurrence extending above 1000 m. However, when clouds below 500 m are simulated in ASR, it produces realistic values of cloud water. Both observed and simulated clouds are dominated by the liquid phase. The lack of observed cloud ice indicates that the ARISE-ASR differences in cloud

water do not result from the parameterization of mixed-phase cloud microphysics. Also, ASR produces little precipitation water, ruling out an overactive cloud water-to-precipitation conversion scheme.

*Can discrepancies between ARISE and ASR be explained by grid-scale disagreements in meteorological conditions and/or errors in the simulated cloud/thermodynamic relationships?*

A number of factors contribute to the low cloud water bias in ASR. With regard to the average meteorological conditions, ASR

is 1.4°C warmer and 0.06 g kg$^{-1}$ drier than ARISE, leading to a ~4% lower relative humidity. The warm temperature bias occurs throughout the boundary layer and is the larger contributor to the lower relative humidity in ASR. In addition, ASR produces clouds across a narrower range of relative humidity values near 100%, while ARISE detected clouds in regions where





the spatially averaged relative humidity was below 50%. This further constricts the opportunity that ASR has to form clouds. Our results also show that the ARISE-ASR differences are attributed primarily to differences in the cloud-thermodynamic

regime relationships, and secondarily (but still importantly) to the warmer and drier simulated thermodynamic regimes. The relative importance of the primary and secondary factors are roughly 2/3 to 1/3. Thus, considering a circumstance where ARISE and ASR have the same thermodynamic conditions, ASR mean total cloud water is ~40% lower than the ARISE observed value.

Considering thermodynamic regimes of temperature, specific humidity, and relative humidity depicts that ARISE observations

show large cloud water values in meteorological regimes of low temperature, low specific humidity, and high relative humidity. ASR, in opposition, shows little relationship between cloud water and temperature, with specific humidity being the only (weak) control. This result is found in both the flight track and Beaufort Sea domain results.

*How are ARISE/ASR comparisons affected by the coverage of ARISE flights?*

Disagreement in the thermodynamic regime-cloud water relationships occurs between the flight track and Beaufort Sea domain

analyses indicate an influence of sampling on the model-observation comparisons. On one hand, the Beaufort Sea domain analysis shows that ASR simulates the high cloud water amounts within the low temperature/low specific humidity regime. This is not evident in the ASR flight track results because the flight path did not intercept this regime, contributing to the low ASR cloud water. Including this low temperature/low specific humidity regime in the full dataset roughly doubles the mean simulated cloud water to about half of the observed value. On the other hand, the effect of the warm and dry (relative humidity)

biases on simulated cloud water are just as severe in the full ASR dataset as in the flight track-only data. Thus, an important lesson from this analysis is that when comparing *in situ* airborne data with model output one should not restrict the comparison to flight track only model output, as expanding the domain isolates omnipresent thermodynamic biases in the reanalysis.

In addition, the full ASR dataset shows a second cloud water maximum within an unobserved high temperature/high specific humidity thermodynamic regime. It is not clear from the ARISE data if the aircraft simply missed sampling this regime, or if

the regime exists.

The effect of limited sampling of ASR along the ARISE flight path estimated with a Monte Carlo random sampling analysis of the full ASR dataset within the Beaufort Sea domain. The results demonstrate that thermodynamic regimes with large cloud water are rare and unlikely to be sampled by chance alone for the number and length of ARISE flights. In several of the regimes, sampling uncertainty obscures the relationships between cloud water and thermodynamic regimes such that it could

not be determined that the ARISE-ASR differences (even large ones) were not caused by sampling uncertainty.

While the ARISE campaign was designed primarily to investigate radiative fluxes in conjunction with satellite overpasses, the random sampling results can also inform the design of future aircraft campaigns intended to test cloud and thermodynamic properties in reanalysis and other atmospheric models. A possible solution to the sampling issues discussed here is to direct aircraft not just to regions where clouds are observed, but also to regions of thermodynamic conditions where models predict

that clouds *should* be, and methodically sample them to build up a sufficient sample size to reduce statistical noise. Moreover, an increased number of flights with a systematic sampling strategy would help reduce sampling uncertainty by raising the



chances of sufficiently sampling the thermodynamic regimes of interest and better capture the variability of cloud conditions occurring in these regimes. This approach will better support one of the key benefits of aircraft campaigns, to sample meteorological properties that satellites cannot, in a more statistically sound manner.

*Data availability.* ARISE data can be found at https://www-air.larc.nasa.gov/cgi-bin/ArcView/arise. ASR version 2 (NCAR, 2017) data can be found at https://doi.org/10.5065/D6X9291B.

*Author contribution.* J. Brant Dodson designed and conducted the large majority of the analyses presented in this manuscript. Patrick C. Taylor assisted in the analysis design, literature review, and various other necessary tasks. Richard Moore provided technical details of ARISE, and David Bromwich the same of ASR. Richard H. Moore, Keith Hines, Kenneth

Thornhill, Chelsea Corr, Bruce Anderson, and Edward Winstead participating in collecting data during ARISE. Joseph R. Bennett provided some analysis and corrections to the original ARISE dataset. J. Brant Dodson prepared the manuscript with contributions from all co-authors.

*Competing interests.* The authors declare that they have no conflict of interest.

*Acknowledgements.* J. Brant Dodson and Patrick C. Taylor were both supported in part by NASA (grant no.

NNH16ZDA001N-NDOA) and the NASA Radiation Sciences Program. P. C. Taylor was also partially supported by the NASA Radiation Budget Science Project. Richard H. Moore and Chelsea A. Corr were supported by NASA Postdoctoral Fellowships. The NASA Langley Aerosol Research Group (LARGE; https://science.larc.nasa.gov/large/), including Richard H. Moore, Kenneth Thornhill, Chelsea A. Corr, Bruce Anderson, and Edward Winstead were funded by the NASA Radiation Sciences Program, managed by Dr. Hal Maring. The participation by D. Bromwich and K. Hines in this research

was funded by ONR grant N00014-18-1-2361.

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

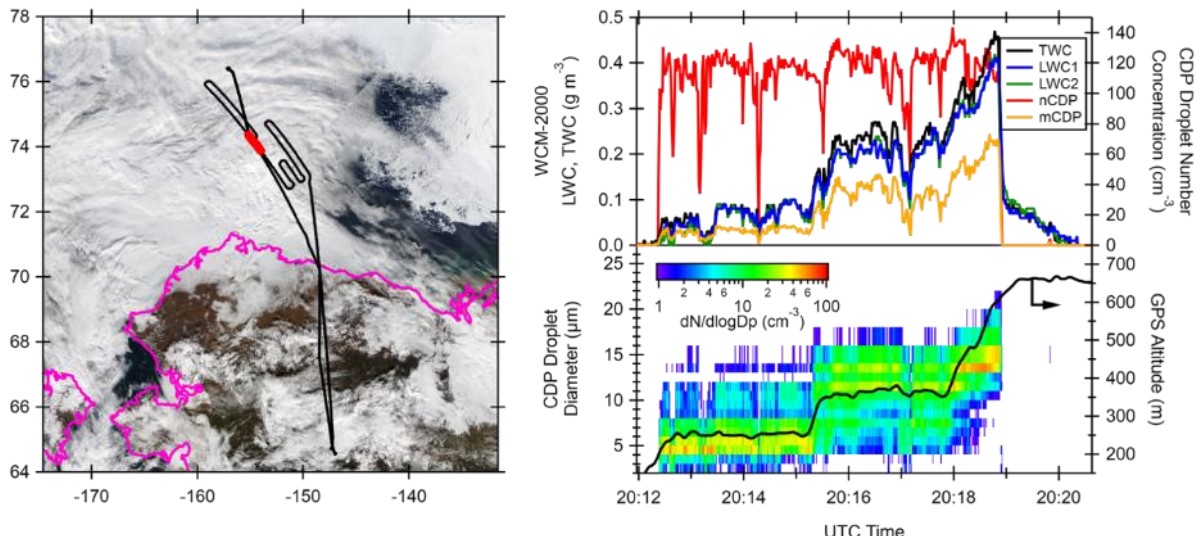

**Figure 1: Example map (left) and timeseries (right) of profile sampling of low-level clouds during ARISE. The black trace in the**
**map indicates the C-130 flight track from Eieleson AFB north to the Beaufort Sea, while the red shading denotes the portion of the flight shown in the timeseries. The timeseries data include CDP measurements of droplet number size distribution (dN/dlogDp), integrated number concentration (nCDP), integrated mass concentration (mCDP; calculated assuming a density of 1 g cm⁻³). Corresponding total water content (TWC) and liquid water content (LWC1, LWC2) mass concentrations measured by the WCM-2000 are also shown. The black trace in the lower right panel is the aircraft GPS altitude.**





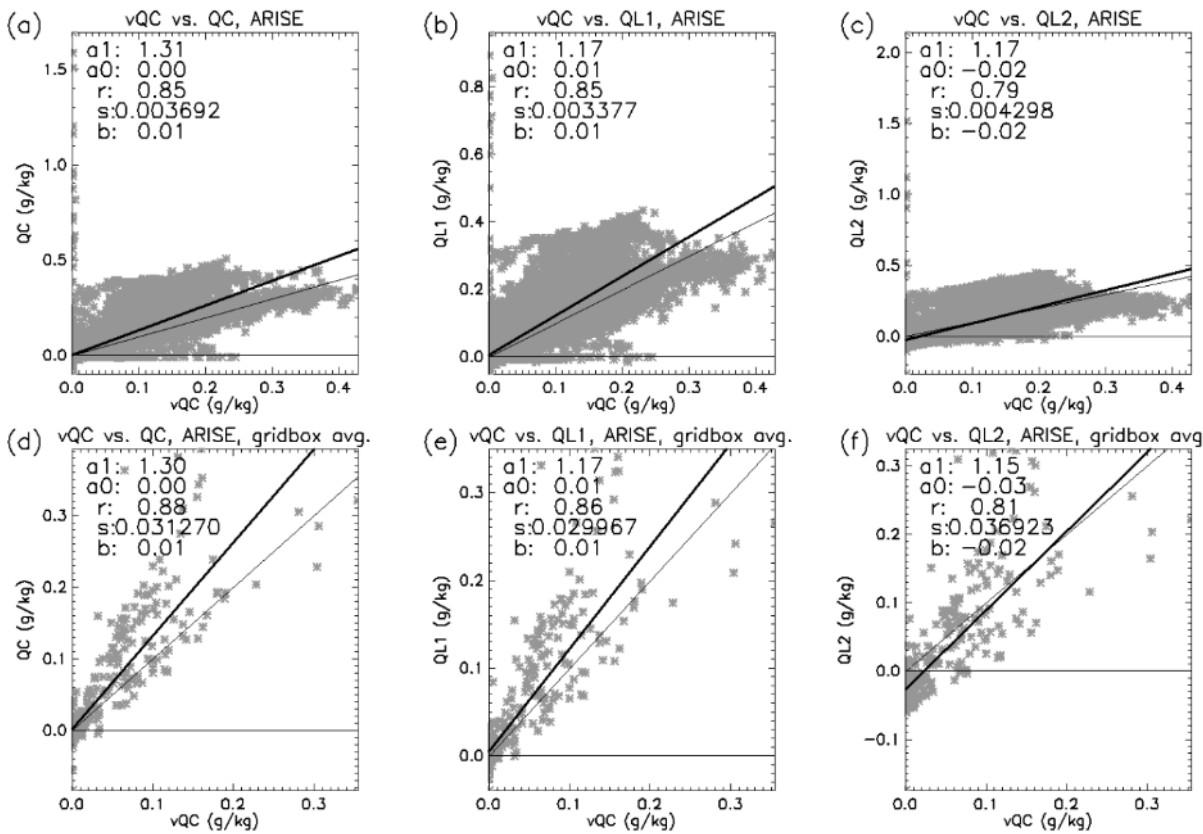


**Figure 2. (a, b, c) Scatterplots of observed cloud water and cloud liquid from the WCM-2000 instrument (y-axis) versus that estimated from the Cloud Droplet Probe (x-axis). vQC is cloud water derived from the Cloud Droplet Probe, QC is total cloud water from WCM-2000, QL1 is cloud liquid from the 2.0 mm wire sensor, and QL2 is cloud water from the 0.5 mm wire. The thick black line is the line of best fit, and the thin black line is the "perfect match" line. Note that the axis values of (a) and (b) are different. Also**
**listed are the regression statistics: slope (a1), y-intercept (a0), correlation (r), standard deviation (s), and bias (b).**

**(d, e, f) Same as (a, b, c), except for data averaged on the ASR grid scale.**

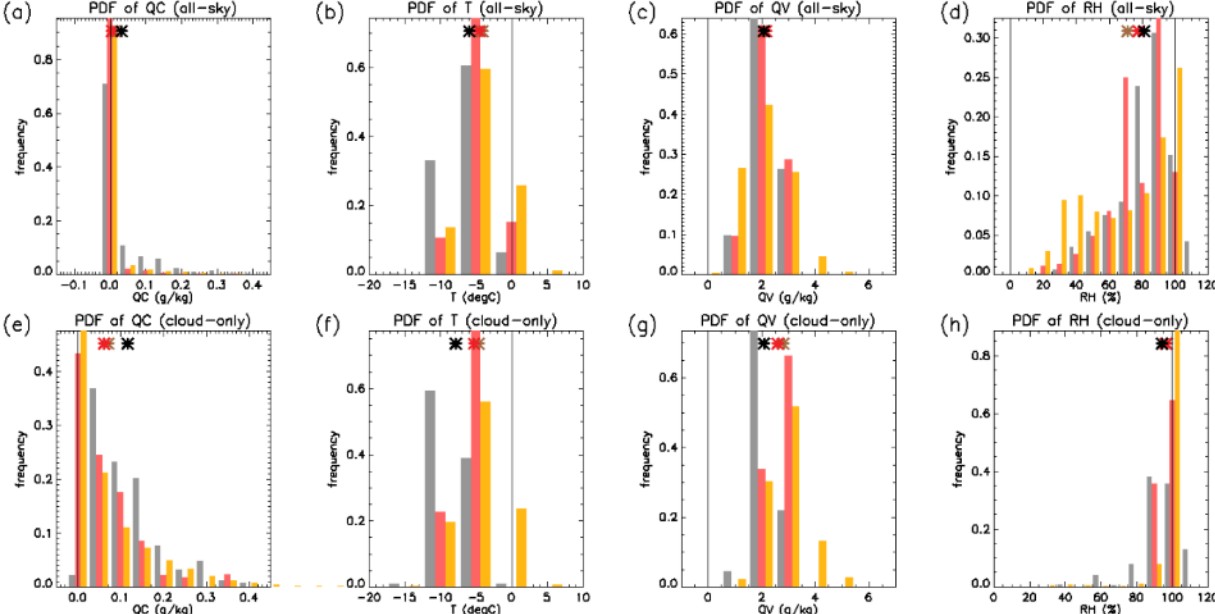

**Figure 3. (a-d) PDFs of cloud and thermodynamic properties of the Arctic environment for ARISE (black), the co-located values of ASR along the ARISE flight path (red), and the ASR values across the entire Beaufort Sea domain (orange). In addition, the mean values are indicated by the asterisks. (a) shows the cloud water (liquid + ice) specific humidity (QC); (b) is temperature (T); (c) is water vapor specific humidity (QV), and (d) is relative humidity (RH).**

**(e-h) Same as (a-d), but for cloudy pixels only (i.e. pixels with mean QC > 0.02 g kg⁻¹).**



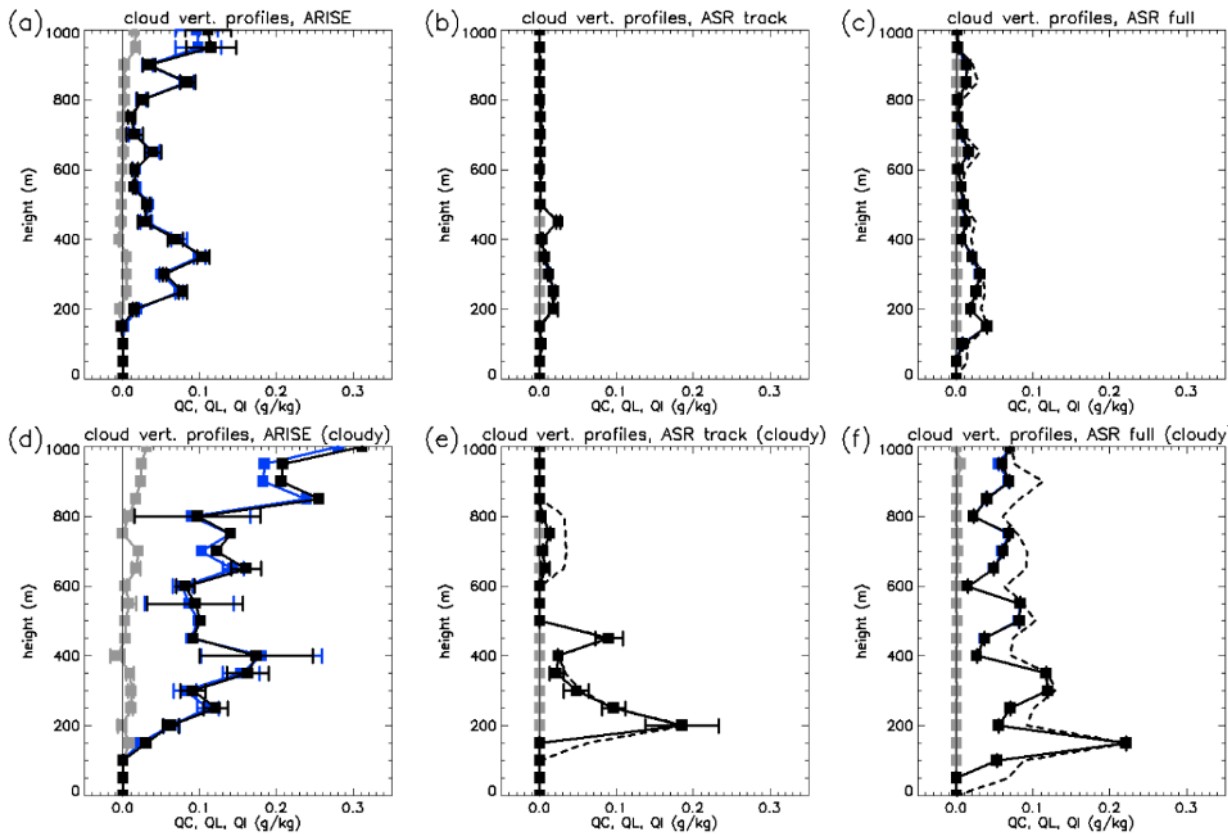

**Figure 4. (a) The vertical profile of cloud water (QC, black), cloud liquid (QL, blue), and cloud ice (QI, gray) for all ARISE data collected from below 1000 m.**

**(b) Same as (a), but for ASR along the ARISE flight tracks. In addition, the dashed black line has precipitation included with the cloud water.**

**(c) Same as (b), but for the full ASR dataset from the Beaufort Seas domain.**

**(d, e, f) Same as (a, b, c), but for cloud-only pixels.**

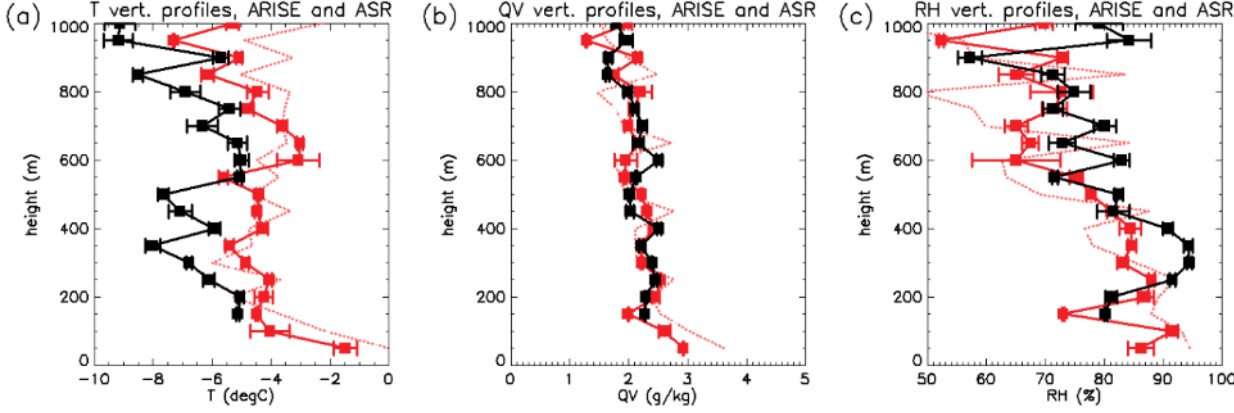


**Figure 5. Vertical profiles of (a) temperature (T), (b) specific humidity (QV), and (c) relative humidity (RH) for ARISE (black) and ASR (red) along the ARISE flight tracks. The whiskers represent the 95% confidence interval using a two –tailed t test. In addition, the vertical profiles from ASR for the full Beaufort Sea domain are shown with the dashed red lines.**



**Figure 6. (a, b, c) Scatterplots of cloud water (QC) versus (a) temperature (T), (b) specific humidity (QV), and (c) relative humidity (RH) for ARISE low-altitude observations. The heavy black line is the linear regression line of best fit.**

**(d, e, f) Same as (a, b, c), but for ASR simulated variables along the ARISE flight path.**

**(g, h, i) Same as (a, b, c), but for ASR simulated variables over the full Beaufort Sea domain.**





**Figure 7. Same as Fig. 6, but for cloudy grid boxes.**





**Figure 8. (a-c)** Mean cloud water specific humidity (QC) for meteorological regimes characterized by temperature (T) and relative humidity (RH). (a) shows ARISE-observed QC averaged by ARISE-observed RH and T; (b) is co-located ASR QC averaged by co-located ASR RH and T; and (c) shows the ASR results for all pixels in the ARISE operational area at all times during the eight days of the ARISE flights (i.e. the population results). Note that gray boxes denote regimes which were not sampled by ARISE (a) or simulated by ASR (b, c).

**(d-f)** Same as (a-d), but for regimes characterized by water vapor specific and relative humidities.

**(g-i)** Same as (a-d), but for regimes characterized by temperature and specific humidity.







**Figure 9. (a-d) Results of the random sampling experiment, which are divided by each thermodynamic regime specified by relative humidity and temperature. (a) shows the chance of a regime being sampled during a single trial set of eight flights; (b) is the typical number of samples collected from each regime during single trial; (c) is the mean absolute error of all trial QC values relative to the population QC values; and (d) shows the standard deviation of the QC values for all trials.**

**(e-h) Same as (a-d), but for regimes characterized by water vapor specific and relative humidities.**

**(i-l) Same as (a-d), but for regimes characterized by temperature and specific humidity.**






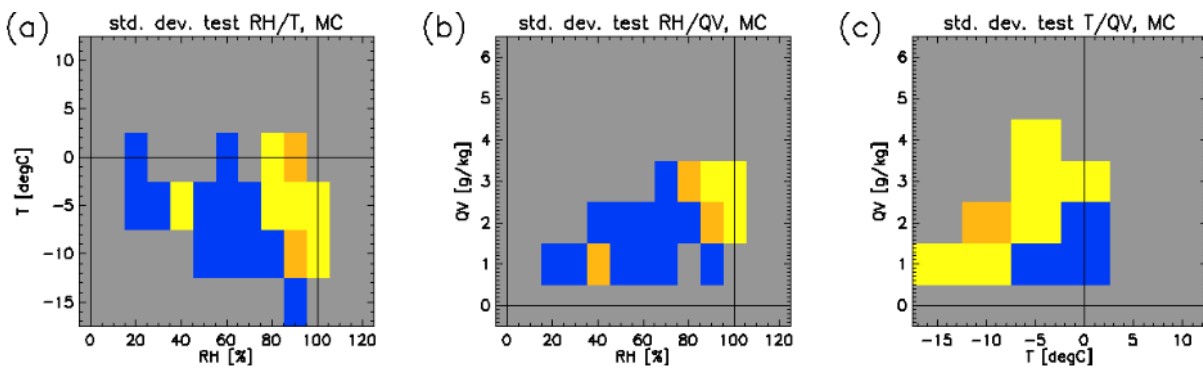

**Figure 10. The closeness of the observed ARISE QC per thermodynamic regime to the simulated QC from the full ASR dataset. The metric is units of standard deviation, where the standard deviations are taken from the random sampling results (Figs. 8d, 8h, and 8l). Yellow colors indicate regimes with QC within one standard deviation, orange is two, and blue is greater than two. Gray are regimes lacking data from ARISE and/or ASR. (a) shows regimes of temperature and relative humidity, (b) of relative humidity and specific humidity, and (c) of temperature and specific humidity.**
