# Peer review of "Evaluation of Simulated Cloud Liquid Water in Low Clouds over the Beaufort Sea in Arctic System Reanalysis using ARISE Airborne In Situ Observations"

_Atmospheric Chemistry and Physics, 2020_

## Referee Comment (RC1) · Anonymous Referee #1 · 30 Dec 2020

This paper presents a comparison of cloud water derived from the Arctic System Re-analysis version 2 (ASR) to measurements made in situ over the Beaufort Sea during the ARISE campaign in September, 2014. The manuscript has three main compo-nents: (1) a comparison between the cloud water properties of ARISE and ASR which accounts for variations in environmental state, (2) an explanation of the observed dif-ferences between the data sets, and similarly (3) an evaluation of the impact limited flight sampling has on the analysis of the ASR cloud water product.

This topic of this paper is highly relevant and fits within the scope of ACP. The scientific reasoning is sound. I appreciate the discussion of how to improve sampling methods

of future airborne field campaigns that aim to study cloud processes. This paper is very well written, and I recommend publication with consideration to a few minor comments.

L39-40 "Through these processes, clouds shape the temperature and sea ice variability and trends in the central Arctic representing a potentially significant climate feedback." This sentence is worded in a confusing manner. Should the second "and" be deleted?

L103 The description of the cloud sampling method would be easier to follow if it is mention that the 2mm wire is used for the calculation of cloud water. This explanation is given in the following paragraph but I don't see the harm in telling the reader this fact on the spot.

L121/L130 Specifically, will ASRv3 have more sophisticated cloud microphysics scheme?

L130 "...with issues such as explored here." Could add the word "those" in front of explored.

L185/L405 Do you know why mixed-phase cloud processes are not occurring in ASR? In explaining the lack of ice, could ice be forming but quickly falling out? That is, I'm unclear about the evidence supporting the notion that mixed-phase cloud processes are not occurring in ASR versus mixed-phase processes just being poorly implemented. Though I admit that this distinction is not terribly important. Also, it's not clear if the use of the term precipitation in the manuscript includes ice and liquid, or only liquid?

L238 Should it be "sizes", and not "sized"? This sentence could be worded in a less confusing manner.

L322 The sentence starting "Implying..." is awkwardly associated with the previous sentence.

L436-437 The first sentence of this paragraph is incomplete.

L555, Fig 1: The left panel needs units or label.

---

## Referee Comment (RC2) · Anonymous Referee #2 · 3 Feb 2021

This manuscript uses airborne in situ observations of low-cloud properties from the ARISE field campaign to evaluate the Arctic System Reanalysis (ASR) product based on thermodynamic regimes of air temperature, relative humidity and specific humidity. The authors find that the ASR product is generally too warm and dry compared to ARISE, which may contribute to biases in the cloud properties, but that the relationship between cloud properties and thermodynamic variables are more important for explaining the difference between the reanalysis data and ARISE. Finally, the authors also determine the impact of the limited sample size of their data on their results. They conclude that sampling noise may influence the comparison between ARISE and ASR and that future comparisons with limited in situ airborne data should not be restricted

to the flight-track.

This is an invaluable study; the evaluation of ASR and the thermodynamic-cloud relationships are important for the improvement of models. The manuscript is well within the scope of ACP. I have some major and minor comments that I would suggest the authors consider:

My main concerns are 1) the interdependence of the thermodynamic variables with one another (they are not independent), 2) that assumption that these thermodynamic variables explain the bulk of the cloud properties and 3) uncertainties in the ARISE dataset that should be described in more detail. Regarding the second point, I am wondering whether cloud condensation nuclei (CCN) could explain some of the large differences in the vertical cloud profiles in ASR compared to ARISE. The authors mention the lack of ice in ASR and maybe the lack of importance of ice-nucleating particles, but there are also CCN. If CCN are important, then this may imply that local sources are important and that comparisons outside the flight track may not be fair comparisons.

Minor comments:

1) Figures 6 and 7: It would be more informative to make those datapoints show the frequency of occurrence of the datapoints using a colour as a third dimension. 2) Lines 320 to 323: should be one sentence. 3) In general, I think the writing can be clarified to state the key points more clearly. Many of the discussions can be distilled to simpler messages.

---

## Author Comment (AC2) · 25 Mar 2021

**Response to Anonymous Referee #2**

(Referee comments are in black; our responses are in blue.)

Thanks very much for recognizing the value of the research, and for the thoughtful comments which helped us to improve upon the original manuscript. We've chosen to address the comments point-by-point below

My main concerns are

1) the interdependence of the thermodynamic variables with one another (they are not independent)…

Certainly the variables are not independent. This was stated in subsection 4.2 in the original manuscript, and to make the issue clearer, we added the following to subsection 3.1:

"Bear in mind that T, QV, and RH are not entirely independent. Indeed, RH is a function of both T and QV, and maximum QV (approximately equilibrium QV) is limited by T. So the three variables should not be interpreted as three separate metrics of the thermodynamic state. Rather, they represent three different related, but distinct, means of characterizing the thermodynamic state."

2) …that assumption that these thermodynamic variables explain the bulk of the cloud properties…

See below for this response.

3) …and uncertainties in the ARISE dataset that should be described in more detail.

The headers for the ARISE data files (found at https://www-air.larc.nasa.gov/) report the uncertainties of QC, T, and RH as about 5% of the measurement, 0.5°C, and 2-5% (percentage of RH, not of the measurement). An uncertainty value for QV is not provided. These uncertainties are for the un-averaged 1 hz measurements, not of the grid box averaged data. The stated cloud water uncertainty may be a bit low, as shown by Fig. 2 and the associated discussion.

Panel E shows that the disagreement between the two sensors can be more than 100% for individual grid boxes. But overall, at larger QC values, CDP tends to report (derived) QC values ~25% lower than WCM-2000. An uncertainty of this size is notable (assuming the CDP is truth), but alone is not enough to explain the 50-75% disagreement between ARISE (from the WCM) and ASR.

The manuscript text in subsections 2.1 and 3.1, and section 6 are now updated to describe these uncertainties.

Regarding the second point, I am wondering whether cloud condensation nuclei (CCN) could explain some of the large differences in the vertical cloud profiles in ASR compared to ARISE. The authors mention the lack of ice in ASR and maybe the lack of importance of ice-nucleating particles, but there are also CCN. If CCN are important, then this may imply that local sources are important and that comparisons outside the flight track may not be fair comparisons.

The analysis was based on meteorological properties that were measured by the aircraft. Of course there are other important factors controlling QC, including circulation (both boundary layer and free troposphere) and aerosols. These factors are not entirely independent. With respect to aerosols, it isn't clear how disagreements in CCN concentration would lead to a 50-75% error in ASR QC. ASR microphysics does not account for CCN (or ice forming nuclei (IFN)), and instead is set to simulate clouds in "typical" conditions (including CCN and IFN concentrations). So any aerosol-related disagreement would have to result from anomalous real-world CCN concentrations causing the Sept 2014 clouds to behave much differently than "typical".

Were there anomalous CCN concentrations over the Beaufort Sea during Sept 2014? There are a few different datasets that could help figure this out. One quick way to roughly estimate aerosol conditions during Sept 2014 is with MERRA2, where monthly mean aerosol data can be accessed via the NASA Worldview tool (https://worldview.earthdata.nasa.gov/). Looking at the "aerosol optical depth" overlay, MERRA2 does not

indicate particularly anomalous AOD conditions during Sept 2014, being greater than that of Sept 2013, but lower than that of Sept 2017, for example. Granted, this is a reanalysis product, not actual measurements, and AOD is not the same thing as CCN concentration. But it suggests that Sept 2014 *likely* did not have substantially "atypical" CCN conditions.

But what if there had been anomalous conditions? Variability in CCN concentration can change the number of cloud droplets, with droplet size varying inversely with number. This can have major effects on certain macrophysical cloud properties such as albedo, but QC? The immediate effect of CCN variability mainly redistributes QC between droplets, so there would need to be secondary mechanisms at work for anomalous CCN to significantly affect QC. It might be helpful to think of some scenarios in which CCN concentration variability can greatly alter QC via secondary mechanisms. One possibility could be that high CCN concentration in the real atmosphere during Sept 2014 suppressed precipitation rates and extended the cloud lifetimes (i.e. the Albrecht effect); if ASR did not account for this, it could produce too much precipitation and desiccate the simulated clouds. But as discussed with Referee #1, there is no clear evidence that ASR has a problem with overactive precipitation formation. Coupled with the lack of evidence for high CCN concentration, this mechanism probably isn't a major factor in our results.

So what other mechanism could there be for aerosols to play in this disagreement? Perhaps there could be some small effects by making cloud droplets more or less susceptible to evaporation from dry air entrainment (as discussed by Ackerman et al. (2004)). But any significant effect of aerosols on QC would probably have to include IFN and mixed-phase processes, not just CCN. And as stated in the paper, there does not appear to be enough QI in ARISE or ASR for mixed-phase processes to be a dominant factor. For these reasons we do not think that CCN can lead to a 50-75% disagreement between ARISE and ASR.

We added a short paragraph at the end of subsection 2.1 which briefly summarizes the main points of this discussion.

Minor comments:
1) Figures 6 and 7: It would be more informative to make those datapoints show the frequency of occurrence of the datapoints using a colour as a third dimension.

We've adjusted Figs. 6 and 7 (and Fig. 2) as suggested.

2) Lines 320 to 323: should be one sentence.

The second sentence was rewritten to be a complete sentence. Combining the two would be a bit of a run-on sentence.

3) In general, I think the writing can be clarified to state the key points more clearly. Many of the discussions can be distilled to simpler messages.

We have edited the manuscript with an eye towards to clarify the key points and distill the discussion into simpler messages.

---

## Author Comment (AC1)

**Response to Anonymous Referee #1**

(Referee comments are in black; our responses are in blue.)

Thanks very much for the compliments and for the constructive comments. We've chosen to address the comments point-by-point below

L39-40 "Through these processes, clouds shape the temperature and sea ice variability and trends in the central Arctic representing a potentially significant climate feedback." This sentence is worded in a confusing manner. Should the second "and" be deleted?

This sentence is now written as "Through these processes, clouds shape long-term trends in both temperature and sea ice variability in the central Arctic, representing a potentially significant climate feedback."

L103 The description of the cloud sampling method would be easier to follow if it is mention that the 2mm wire is used for the calculation of cloud water. This explanation is given in the following paragraph but I don't see the harm in telling the reader this fact on the spot.

done

L121/L130 Specifically, will ASRv3 have more sophisticated cloud microphysics scheme?

Yes. The earlier ASR used the Goddard microphysics scheme of Tao et al. (2003). Many of the details of the microphysics are given by Tao and Simpson (1989). The next version of the ASR will use a more advanced microphysics scheme with a demonstrated ability to model polar clouds, perhaps the two-moment P3 scheme of Morrison and Milbrandt (2015).

L130 "...with issues such as explored here." Could add the word "those" in front of explored.

done

L185/L405 Do you know why mixed-phase cloud processes are not occurring in ASR? In explaining the lack of ice, could ice be forming but quickly falling out? That is, I'm unclear about the evidence supporting the notion that mixed-phase cloud processes are not occurring in ASR versus mixed-phase processes just being poorly implemented. Though I admit that this distinction is not terribly important. Also, it's not clear if the use of the term precipitation in the manuscript includes ice and liquid, or only liquid?

This is a complex topic to address, and one that is not necessarily entirely in the scope of the paper. There are a few things to consider:

1. Even if ASR simulated realistic amounts of cloud ice, there was not much cloud ice observed by ARISE, so there would be few opportunities for simulated mixed-phase processes to occur.

2. The purpose of the dashed lines in Figs. 4e and 4f are to show that simulated cloud ice is not simply converting into snow. Yes, by "precipitation" we mean both liquid and ice forms – in fact, snow is the main precipitation species, with rain and graupel being almost nonexistent.

3. If the concern is that cloud ice is converted to snow and then quickly falls out, perhaps because of an overaggressive fall speed parameter, we might expect to see that the precipitation vertical profile is shifted lower in the atmosphere with respect to the cloud water, with the rapidly-falling snow accumulating as it descends through the cloud layer. But this is not observed in the data.

Below are mean vertical profiles of snow (QS, gray), rain (QR, blue), graupel (QG, brown), and QC (dashed black) for the full ASR dataset (left) and the full cloud-only set (right). We set the vertical

coordinate .to pressure to better display the reanalysis data. QS clearly increases with height, the opposite of QC. This does not resemble the profile we might expect from an unrealistic conversion rate from QC to QS, and QS fall speed.

[Figure]

4. ASR has a warm bias over ARISE, particularly in cloudy conditions (about 3°C, see Fig. 3f). While this warm bias does not entirely eliminate all possible conditions for significant cloud ice formation, it makes ice formation less likely.

5. In certain conditions when both ARISE and ASR observe/produce cloud ice, it is possible that ARISE has greater QI because of secondary ice production processes, namely ice crystal splintering, that ASR does not simulate. This was observed in other field campaigns near Antarctica, and in conditions with T > -10°C, and can boost QI by 1-3 orders of magnitude greater than predicted without this process (Grosvenor et al., 2012; O'Shea et al., 2017). Perhaps this occurred on occasion during ARISE, where T was often greater than -10°C. But the main problem with this argument is that ice splintering is a *secondary* process, so even if it was perfectly accounted for in ASR, it would have no effect when ASR fails to produce *any* QI, which is very common. So it isn't clear how significant this issue is. It could be an interesting topic for future research.

Grosvenor, D.P., Choularton, T.W., Lachlan-Cope, T., Gallagher, M., Crosier, J., Bower, K.N., Ladkin, R.S., & Dorsey, J.R. (2012), In-situ aircraft observations of ice concentrations within clouds over the Antarctic Peninsula and Larsen Ice Shelf, Atmos. Chem. Phys., 12, 11,275-11,294, doi:10.5194/acp-12-11275-2012.

O'Shea, S.J., Choularton, T.W., Flynn, M., Bower, K.N., Gallagher, M., Crosier, J., Williams, P., Crawford, I., Fleming, Z.L., Listowski, C., Kirchgaessner, A., Ladkin, R.S., & Lachlan-Cope, T. (2017), In situ measurements of cloud microphysics and aerosol over coastal Antarctica during the MAC campaign, Atmos. Chem. Phys., 17, 13,049-13,070, doi:10.5194/acp-17-13049-2017.

L238 Should it be "sizes", and not "sized"? This sentence could be worded in a less confusing manner.

done

L322 The sentence starting "Implying..." is awkwardly associated with the previous sentence.

The sentence now starts with "This implies…"

L436-437 The first sentence of this paragraph is incomplete.

We inserted an "is" to give the sentence a verb.

L555, Fig 1: The left panel needs units or label.

done